# Extreme warming challenges sentinel status of kelp forests as indicators of climate change

Daniel Reed[1], Libe Washburn[1,2], Andrew Rassweiler[3], Robert Miller[1], Tom Bell[4] & Shannon Harrer[1]

The desire to use sentinel species as early warning indicators of impending climate change effects on entire ecosystems is attractive, but we need to verify that such approaches have sound biological foundations. A recent large-scale warming event in the North Pacific Ocean of unprecedented magnitude and duration allowed us to evaluate the sentinel status of giant kelp, a coastal foundation species that thrives in cold, nutrient-rich waters and is considered sensitive to warming. Here, we show that giant kelp and the majority of species that associate with it did not presage ecosystem effects of extreme warming off southern California despite giant kelp's expected vulnerability. Our results challenge the general perception that kelp-dominated systems are highly vulnerable to extreme warming events and expose the more general risk of relying on supposed sentinel species that are assumed to be very sensitive to climate change.

[1] Marine Science Institute, University of California, Santa Barbara, California 93106, USA. [2] Department of Geography, University of California, Santa Barbara, California 93106, USA. [3] Department of Biological Science, Florida State University, Tallahassee, Florida 32304, USA. [4] Earth Research Institute, University of California, Santa Barbara, California 93106, USA. Correspondence and requests for materials should be addressed to D.R. (email: dan.reed@lifesci.ucsb.edu).

The ecological effects of global warming are expected to be large[1,2], but are proving difficult and costly to measure. This has led to a growing interest in using sentinel species as early warning indicators of impending climate change effects on entire ecosystems[3,4]. Identifying sentinel species most likely to be early indicators of climate change would seem a logical way to direct limited monitoring and conservation resources[5]. If such sentinel species were also foundation species on which other species depend, then their decline could destabilize and collapse entire ecosystems[6,7]. Coastal marine ecosystems supported by foundation species in temperate seas (for example, kelp forests, seagrass meadows and salt marshes) are reported to be particularly sensitive to ocean warming and low-nutrient availability associated with climate change[8], and thus are expected to have been greatly affected by the extraordinary warming that recently occurred off the Pacific coast of North America[9].

The ocean warming began in late 2013 when unusually warm waters extending to depths exceeding 100 m were observed in the central North Pacific Ocean with temperature anomalies exceeding three s.d. ($\sim$ 3 °C) above the climate record from 1982–2014 (refs 9–11). The warming resulted from higher than average sea-level atmospheric pressure that in turn led to below average heat loss from the ocean, weaker winds and less cold-water advection in the upper ocean[12]. The warm water 'blob' moved eastward, and by October 2014 spread to fill the coastal domain from the Gulf of Alaska to the tip of Baja California, Mexico[12,13]. This phenomenon was followed by severe El Niño conditions, which prolonged the warming through 2015. These changes were unprecedented, possibly since the early 1900s[9,11,12] and have been linked to ecological changes in pelagic ecosystems such as reduced nutrients and phytoplankton primary productivity, the appearance of unusual species of zooplankton and fish, and increased mortality in marine birds and mammals[9,13].

Time series data of giant kelp forest ecosystems in southern California collected by the Santa Barbara Coastal Long Term Ecological research program provided us with a unique opportunity to examine the effects of the unprecedented warming on an important coastal foundation species, the giant kelp *Macrocystis pyrifera*, and its associated ecosystem, and to assess the value of kelp forests as sentinels for detecting early signs of climate related impacts. Like most kelps, the growth and survival of *Macrocystis* are considered highly vulnerable to prolonged conditions of warm, nutrient-poor water[14,15] and such conditions have been implicated as the cause for massive regional declines[16,17]. The demography and physiology of *Macrocystis* make it potentially well suited as a climate sentinel because of its rapid growth (2% dry mass per day)[18], high biomass turnover (6–7 times per year)[18] and relatively high nitrogen demand ($>1 \, \mu mol \, l^{-1}$ needed to sustain growth)[19]; yet, limited capacity for internal nitrogen storage ($<3$ weeks)[20] should cause its biomass to respond quickly to prolonged unfavourable growing conditions associated with climate change.

Surprisingly, fluctuations in the biomass of giant kelp and much of its associated biota remained within historical ranges despite 24 months of above average temperatures and below average nutrients. The resilience of giant kelp to the unprecedented warming not only highlights the limitations in our understanding of the ecology of kelp-dominated ecosystems, but also questions their general use as early indicators of climate change.

## Results

**Oceanographic changes.** We observed the first signs of prolonged warming of bottom waters within kelp forests of the Santa Barbara Channel in late December 2013 when temperature anomalies turned conspicuously positive (Fig. 1a). Positive monthly temperature anomalies persisted through December 2015 with daily deviations as high as $+5.8$ °C and monthly deviations averaging as much as $+4.6$ °C. The unprecedented extent of the warming is evidenced by the occurrence and duration of unusually warm water events in 2014–2015 compared with earlier in the time series (Fig. 1b). Fifty-eight per cent of the days during 2014 and 2015 (423 of 730 days) met the criteria of a marine heatwave (that is, five or more consecutive days when the mean temperature is warmer than the 90th percentile)[21] compared with only 3.4% (162 of 4,748 days) during the previous 13 years. Moreover, the duration of heatwaves averaged three times longer during 2014–2015 (mean = 39 days,

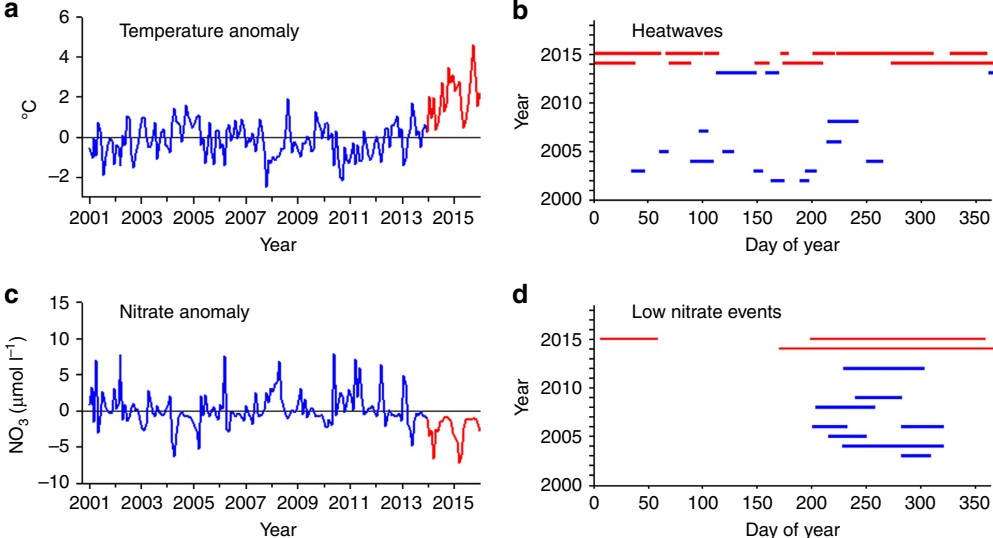

**Figure 1 | Anomalies in ocean temperature and nitrate in the Santa Barbara Channel.** Monthly anomalies in (**a**) bottom temperature at 7–10 m depth and (**c**) bottom nitrate concentrations at 7–10 m depth. Occurrence and duration of (**b**) heatwaves defined as five or more consecutive days when the mean bottom temperature was warmer than the 90th percentile and (**d**) low nitrate events lasting at least 21 days with mean daily bottom nitrate concentrations $<1 \, \mu mol \, l^{-1}$. The anomalously warm years of 2014–2015 are shown in red.

maximum = 151 days, n = 11) compared with 2001–2013 (mean = 12 days, maximum = 30 days, n = 13).

Anomalies in bottom-water nitrate concentrations were negative in every month during 2014 and 2015 (Fig. 1c). Moreover, 2014–2015 was characterized by three prolonged events with critically low nitrate (that is, < 1 µmol l$^{-1}$) lasting 50, 156 and 195 days, respectively, which collectively encompassed 54% of the 2-year period (Fig. 1d). In contrast, events of < 1 µmol l$^{-1}$ lasting at least 21 days (= maximum duration that growth in giant kelp can be sustained on internal nitrogen reserves[20]) were typically much shorter in duration in 2001–2013, and collectively accounted for only 7% of the days during this period.

Large storm waves tear out giant kelp and greatly alter kelp forest communities[14,15]. Large winter swells coincided with the warm, nutrient-poor water associated with the severe El Niños of 1982–1983 and 1997–1998 (refs 22,23), which made it difficult to disentangle the effects of ocean warming and low nutrients on kelp forests from those caused by wave disturbance[16]. Unlike ocean temperature and nutrients, the wave climate in the Santa Barbara Channel during 2014 and 2015 was largely unremarkable. The largest monthly anomaly in maximum significant wave height during the recent warm period was 0.54 m in October 2014 compared with 1.2 m in March 1983 and 2.3 m in February 1998 (Fig. 2a). Moreover, larger swells were recorded at our kelp forest study sites during the cool years of 2006 and 2008 than in the anomalously warm years of 2014 and 2015 (Fig. 2b). Thus, unlike during previous warming episodes of the latter twentieth century, anomalous disturbance from waves was not a confounding factor during the extreme warming in 2014–2015.

**Ecological responses**. The community structure of kelp forests in the region varied substantially throughout the 15-year time series, but surprisingly, large responses to the unprecedented warm, nutrient-poor conditions were not obvious for most components of the community. In particular, we did not detect a dramatic response by giant kelp. Instead, we observed considerable inter-annual variability in its summer biomass with values for 2014 and 2015 within or slightly below the range of values recorded during the cooler years of the time series (Fig. 3a). Overall, kelp biomass decreased significantly during the 15-year record, but this decline was unrelated to anomalies in temperature in the previous year (Table 1, Supplementary Fig. 1a). Understory algae that grow on the sea floor beneath the canopy of giant kelp, and mobile reef fishes were even more unresponsive to the unprecedented warming, as their biomass did not vary significantly over time and remained well within the range of values observed during previous years (Fig. 3b,f; Table 1, Supplementary Fig. 1b,f). Moreover, multivariate analyses showed that the algal and fish assemblages did not differ appreciably between the cool and warm periods (Supplementary Fig. 2a,c). Instead, the species composition of understory algae shifted gradually through time while that of reef fishes exhibited no consistent trend over the record.

Benthic sessile invertebrates that feed on plankton and other suspended matter in the water column constitute the largest consumer group in the kelp forest. Their biomass decreased significantly over the study period (Fig. 3c), but their decline was independent of the anomalously warm conditions (Table 1, Supplementary Fig. 1c). The species assemblage of sessile invertebrates changed gradually over the course of the time series and showed a modest (21.3%) dissimilarity between the cool (2001–2013) and warm (2014–2015) periods (Supplementary Fig. 2b, Supplementary Table 1).

The biomass of sea urchins, the most important grazers in kelp forests worldwide[24], showed no consistent trend over time (Fig. 3d, Table 1). Moreover, neither the biomass of giant kelp nor understory algae explained any of the year-to-year variation in sea urchin biomass (neither variable met the 0.15 significance level for entry into a stepwise multiple regression model regardless of whether data for kelp and understory algae were derived from the same year as sea urchins or the previous year). Instead sea urchin biomass was negatively correlated with anomalies in bottom temperature (Table 1, Supplementary Fig. 1d) Sea urchin biomass reached its lowest level in 2015 when it declined by 50% from 2014. This abrupt decline is consistent with a rapid response to extreme warming that occurred during the previous 12 months. Indeed, our observations suggested that this abrupt decline resulted from an infectious disease (as evidenced by visible lesions on the epidermis) that is known to spread rapidly during warm periods[25]. The absence of a large positive response by giant kelp and understory algae coinciding with this decrease in sea urchin biomass, and the generally poor correlation between sea urchins and their algal food supply throughout the time series, emphasizes the complexity of biotic and abiotic interactions in kelp-dominated systems[26].

The most dramatic response to warming was observed in sea stars, an important group of invertebrate predators. Unlike other components of the kelp forest community, sea stars were increasing before the onset of warming in 2014, which accounted for an overall increasing trend during the time series (Fig. 3e, Table 1). However, in 2014 an epidemic outbreak of wasting disease led to mass mortalities of at least 20 species of sea stars from Alaska to Mexico[27]. Sea star biomass in the kelp forests off Santa Barbara declined to near zero by 2015. Museum specimens show that the densovirus linked to this disease has been present on the Pacific coast of North America for at least 72 years[27] and the anomalously warm water temperatures have been

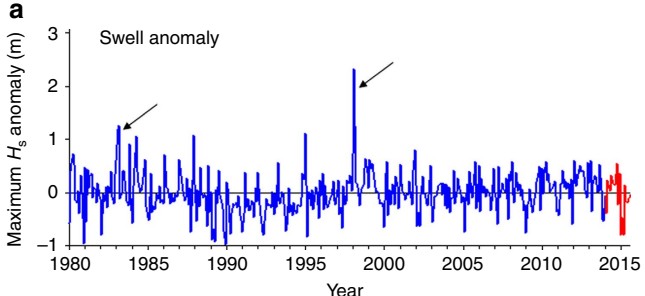

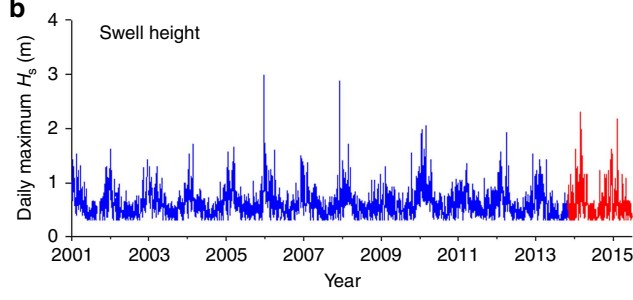

**Figure 2 | Swell height data for the Santa Barbara Channel.** (**a**) Monthly anomaly in the maximum significant wave height (max $H_s$) measured from an offshore buoy for the period 1980–2015 and (**b**) Modelled mean daily values of max $H_s$ averaged across the nine study sites for the period 2001–2015. The anomalously warm years of 2014–2015 are shown in red. Black arrows in (**a**) denote the 1982–83 and 1997–98 El Niños.

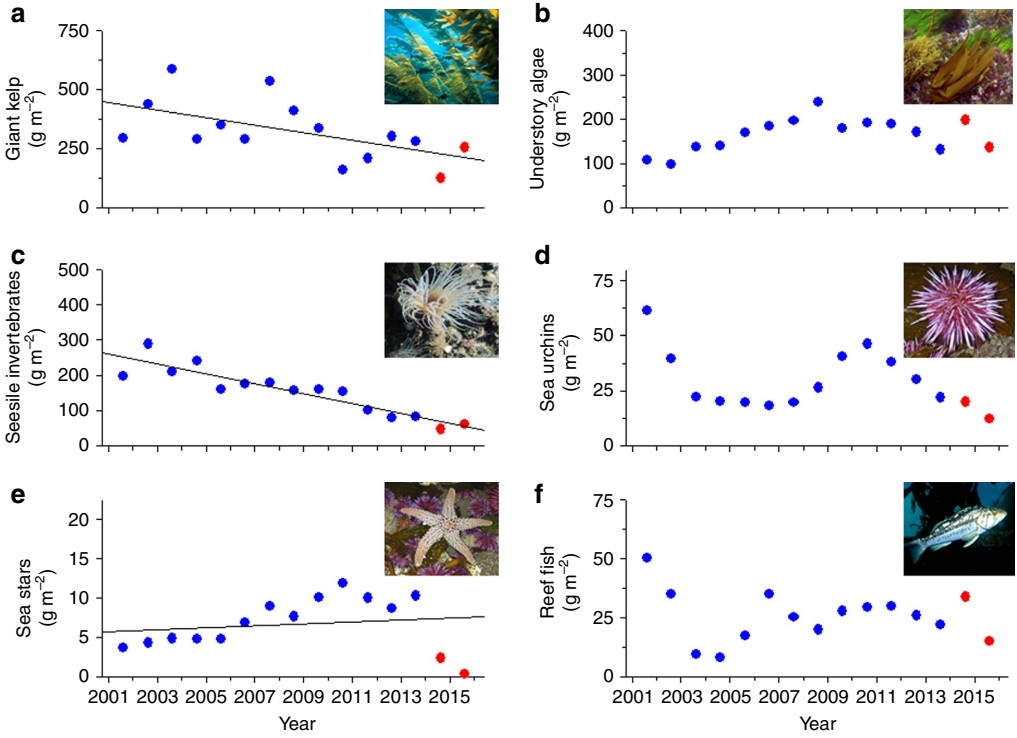

**Figure 3 | Ecosystem responses to extreme warming in the Santa Barbara Channel.** Temporal trends in the biomass of the major groups of kelp forest primary producers and consumers prior to (2001–2013, blue) and during (2014–2015, red) the anomalous warm period. (**a**) giant kelp, (**b**) understory algae, (**c**) sessile suspension feeding invertebrates, (**d**) sea urchins, (**e**) sea stars and (**f**) reef fish. Biomass values are annual means measured in summer averaged over nine sites and are in units of decalcified dry mass, except reef fish which are in units of dry mass. Regression lines are provided for cases where year contributed to a significant model (Table 1).

---

**Table 1 | Variation in ecological responses attributed to oceanographic anomalies.**

| Ecological response | Temperature | Year | Adj model $R^2$ | P value |
|---|---|---|---|---|
| Giant kelp | –* | 0.322 (−) | 0.269 | **0.028** |
| Understory algae | 0.141 (−) | 0.156 (+) | 0.181 | 0.120 |
| Sessile invertebrates | –* | 0.844 (−) | 0.832 | **<0.001** |
| Sea urchins | 0.351 (−) | –* | 0.3101 | **0.017** |
| Sea stars | 0.272 (−) | 0.230 (+) | 0.418 | **0.015** |
| Reef fish | –* | –* | –* | >0.15 |

Results of stepwise multiple regression analyses showing the amount of variation in the summer biomass of different functional groups of kelp forest species (that is, ecological response variables) explained by year and anomalies in bottom temperature for the preceding 12 months. Numbers reported for the independent variables (temperature and year) are partial $R^2$ values; the direction of the relationship involving each independent variable is shown in parentheses. * indicates that the independent variable did not meet the 0.15 significance level for entry into the model. Model P values <0.05 are shown in bold.

---

implicated in increasing its spread and the mortality rate in sea stars[28]. This implication is consistent with the negative relationship that we observed between sea star biomass and bottom temperature anomalies (Table 1, Supplementary Fig. 2e). So far the mass mortalities of sea stars have not led to corresponding increases in their prey. The biomass of piddocks and shelled gastropods, favourite prey of sea stars, did not change appreciably in the 2 years following the disappearance of sea stars (Supplementary Fig. 3), although increases in these longer-lived animals may be expected in the future if sea stars do not recover.

**Discussion**

Giant kelp populations are naturally quite variable, so changes in kelp forests can easily be misinterpreted unless viewed in the context of long-term data[29]. Longer and more spatially comprehensive records of giant kelp and sea surface temperature (SST) obtained from satellite imagery confirm the

patterns documented by divers and moored sensors at our study sites. Large seasonal and interannual fluctuations in giant kelp biomass characterized all of southern California from 1984–2015 (Fig. 4a). However, negative-kelp anomalies during 2014–2015 from Santa Barbara to the Mexican border in San Diego were within the range observed during the 32-year time series despite unprecedented positive anomalies in ocean temperatures during this period (Fig. 4b). This result is quite surprising given giant kelp's reported sensitivity to such conditions[16,17] and stands in contrast to reports from other kelp systems showing sharp declines in response to warming conditions[30–32]. The location of our study within giant kelp's range offers one possible explanation for the difference; southern California, and Santa Barbara in particular, does not represent either the geographical or thermal limit for the species in the northeast Pacific[14,15] although San Diego is on average 1–4 °C warmer than Santa Barbara and closer to giant kelp's thermal limit (Supplementary Fig. 4). Nevertheless, giant kelp populations are highly adapted to

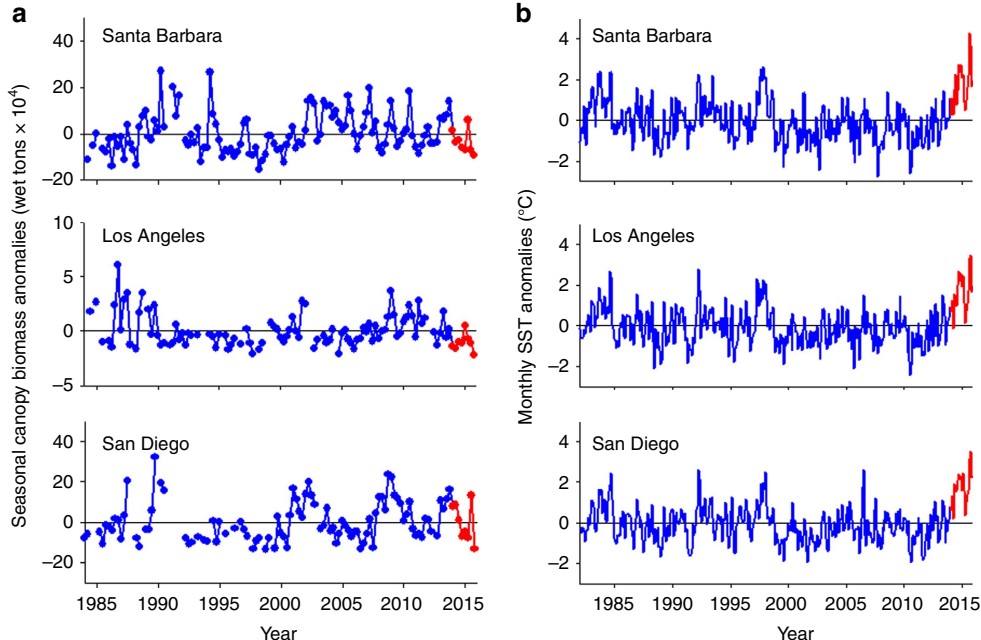

**Figure 4 | Regional trends in giant kelp biomass and SST.** (**a**) Seasonal canopy anomalies of giant kelp biomass estimated from Landsat 5 Thematic Mapper and Landsat 7 Enhanced Thematic Mapper for 1984–2015. (**b**) Monthly SST anomalies from National Climatic Data Center Optimal Interpolation Sea Surface Temperature data for 1984–2015. Data are shown for the counties of Santa Barbara, Los Angeles and San Diego, which span the 450 km coastline of southern California and ~2° latitude (34.5 N in Santa Barbara and 32.2 N in San Diego). Data for 2014–2015 are shown in red.

local nutrient conditions[33] and the potential for rapid dispersal from distant populations that are better adapted to sudden changes in local conditions is limited[34]. Thus, similar to that shown for other habitat forming seaweeds[35], we would expect giant kelp populations far from the range edge to be equally sensitive to large and sustained anomalies in temperature and nutrients such as those associated with the recent warming event.

Anthropogenic climate change is predicted to cause the extinction of thousands of species over the next century[1,2] so understanding the responses of species to climate change is perhaps the most pressing issue facing ecologists today. Anomalous environmental conditions offer exceptional opportunities to test our expectations for ecosystem responses to climate change. The lack of an expected response to an extreme event, as in our study, reveals our limited ability to predict responses to longer term changes. Such limits on our forecasting ability underscore the key role of long-term data in detecting the effects of climate change on ecosystems as they occur[29,36], prompting calls for the need to identify and monitor sentinel species[3–5]. Indeed, given the cost of collecting long-term data, efforts to focus research on the most sensitive and vulnerable species seem wise. However, our results expose the potential dangers of relying on easily measured iconic species with supposed sensitive traits as sentinels for evaluating ecosystem responses to climate change. We suggest that in the absence of an ability to reliably predict sentinel species, broader spectrum ecological monitoring is necessary to inform us of the status of biodiversity in the face of climate change.

## Methods

**Study region.** We collected oceanographic and ecological data from 2001–2015 at nine kelp forest sites located at 7–10 m depth along an 80 km stretch of the mainland coast of the Santa Barbara Channel in southern California, USA (Supplementary Fig. 5).

**Oceanographic data.** Sea temperature was measured every 10 min using loggers (Stowaway Onset tidbits, accuracy ± 0.2 °C; Onset Computer, Bourne,

Massachusetts, USA) fastened to the bottom at each site. Longer records of ocean temperature from 1982–2015 were obtained for the Santa Barbara region from the National Climatic Data Center Optimal Interpolation Sea Surface Temperature data. These data combine measurements from several sources including ship, buoys and Advanced Very High Resolution Radiometer satellite images from January 1982–December 2015 to produce a daily averaged data set with a resolution of 0.25 °C. The values shown in Fig. 4b represent the mean of the pixels that encompass the coastal waters offshore of Santa Barbara, Los Angeles and San Diego counties.

The nutrient considered to most often limit primary production in the Santa Barbara Channel and elsewhere in southern California is nitrogen in the form of nitrate, which can be predicted from temperature[37]. We modelled nitrate concentration at our sites as a function of bottom temperature using the exponential relationship developed by Fram *et al.*[38] for the Santa Barbara Channel.

We used the maximum value of the significant wave height ($H_s$) during each month of the 15-year time series as a measure of wave disturbance. $H_s$ represents the mean of the largest one-third of the waves recorded during a 30-min sampling period and the maximum $H_s$ has been shown to be significantly correlated with the amount of kelp biomass loss[18,39,40]. $H_s$ data for the period 2001–2015 were obtained from the Coastal Data Information Program's nowcast wave-propagation model for locations nearest to each of the nine kelp forest sites[40]. The model provides hourly estimates of $H_s$ along a 10 m isobath at an 800 m longshore resolution. Longer term records of $H_s$ (1980–2015) used to compare the wave climate during our study with that during the severe El Niños of the later 20th century were obtained from the Wave Information Studies hindcast model nearest the location of the Harvest platform buoy (Station 46218) located offshore in the Santa Barbara Channel (34.454 N 120.782 W). The Harvest buoy provided additional $H_s$ data from 2012–2015.

Monthly values of bottom temperature, bottom nitrate and maximum $H_s$ were calculated for each of the nine kelp forest sites and were averaged to produce a regional mean value for each day of the time series. Regional daily means were averaged to produce regional monthly means. Regional monthly anomalies for bottom temperature and bottom nitrate were calculated as the difference between the regional mean value for a given month in a given year and the 15-year average for that month. Monthly anomalies for maximum $H_s$ used to compare the wave climate during the recent warming in 2014–2015 with that during the 1982–1983 and 1997–1998 El Niños were calculated as the difference between the mean obtained using modelled hindcast and buoy data from Harvest Platform for a given month in a given year and the 36-year average (1980–2015) for that month. To further characterize the oceanographic conditions at our study sites we applied Hobday *et al.*'s hierarchical approach for defining marine heatwaves[21] to the bottom temperature data and calculated the occurrence and duration of heatwaves in each year of the time series. We estimated low nitrate events as periods when a mean daily nitrate concentration of 1 µmol l$^{-1}$ persisted for at least 21 days. The rationale for this definition is based on previous studies that concluded

concentrations of nitrate $< 1 \, \mu mol \, l^{-1}$ are not sufficient to support normal growth in *Macrocystis*[19] and its internal nitrogen reserves become exhausted within 21 days without any external supplies[20].

**Ecological data.** Annual community surveys of marine flora and fauna were performed by divers using SCUBA in late July to early August each year. Sampling was done within 34 permanently located $40 \, m \times 2 \, m$ plots distributed among the nine kelp forest sites. The density and size of giant kelp, large solitary understory algae, large mobile invertebrates and reef-associated fishes were measured within each plot (Supplementary Table 2). Smaller mobile invertebrates and solitary algae were counted within six $1 \, m^2$ quadrats placed at 8 m intervals along the 40 m axis of each plot. The per cent cover of sessile invertebrates and understory algae that are impractical to count as individuals was measured at 80 uniformly spaced points within each plot. Data on size and abundance of each species counted in the surveys were converted into biomass using species specific relationships developed for algae[41,42], invertebrates[43] and fish[44–47] in the region. Mean biomass values for the different taxonomic groups examined were calculated for each of the nine sites and then averaged to produce a regional mean value for each year.

Longer more spatially comprehensive records of giant kelp canopy biomass were estimated at 30-m resolution from 1984–2015 using multispectral Landsat 5 and 7 Thematic Mapper satellite imagery following procedures developed by Cavanaugh *et al.*[48]. Kelp canopy biomass was estimated using the observed relationship between diver-estimated kelp canopy biomass at our study sites and Landsat pixel kelp fraction. Cloud-free imagery allowed kelp biomass to be estimated approximately monthly. Seasonal canopy biomass determinations for each $30 \, m \times 30 \, m$ pixel along the mainland coast of the counties of Santa Barbara, Los Angeles and San Diego Channel were summed to produce a value of kelp canopy biomass for each county, for each season of the 32-year time series.

**Statistical analyses.** The primary focus of our study was on detecting regional responses to a large-scale warming event. Therefore, our analyses involved models using regional means averaged across the nine sites rather than estimates of the mean and variance of variables from the time series of individual sites (for example, the regional mean and variance in the relationship between local kelp biomass and local temperature that would be derived from a mixed effects model). Thus we evaluated the amount of variation in the regional summer biomass of the different functional groups of kelp forest species that was explained by anomalous warming using stepwise multiple regression with a threshold level of significance $= 0.15$ for entry into the model (SAS 9.4, SAS Institute Inc., Cary, NC, USA). Annual anomalies in bottom temperature for the 12-month period preceding the summer measurements of biomass were used as an independent variable in the regression analysis of each functional group. Year was included as an additional independent continuous variable to account for temporal trends not explained by anomalous oceanographic conditions. Multi-collinearity of the two independent variables was low in all cases; condition indices were $< 26$ and variance inflation factors $< 2$ for all analyses.

Differences in the species structure of understory algae, sessile invertebrates and reef fishes before and after the extreme warming were analysed using multidimensional scaling[49] and permutation-based analysis of variance using Type III sums of squares[50] followed by similarity percentages (SIMPER) to decompose the contributions of individual species leading to dissimilarities (Bray–Curtis) between the cool (2001–2013) and warm (2014–2015) periods. While multidimensional scaling plots provide an intuitive means of evaluating differences in species assemblages among years, the statistical significance of differences between cool and warm periods as determined by permutation-based analysis of variance should be viewed with caution due to the unbalanced replication of years between the two periods.

**Data availability.** Data collected by SBC LTER that support the findings of this study are available with the following identifiers: kelp forest species biomass doi:10.6073/pasta/449a810468445ef1ce60769466cd1fad; bottom temperature: doi:10.6073/pasta/c3ec3193ba97e7670b532cfbbe1632f9; Landsat kelp biomass (1984–2011) doi:10.6073/pasta/329658f19d5e61dda0be5ee883cd1c41; Landsat kelp biomass (2012–2015) available from authors. Access to these data require users to accept SBC LTER's data use agreement.

SST and swell height data used in this study are publically available at: SST http://www.ncdc.noaa.gov/oisst/; CDIP swell height http://cdip.ucsd.edu/; Wave Information Studies hindcast model http://wis.usace.army.mil/; Platform Harvest swell height http://www.ndbc.noaa.gov/station_history.php?station=46218.

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

## Acknowledgements

The research was supported by the U.S. National Science Foundation's Long Term Ecological Research program (OCE 9982105, 0620276 & 1232779) and by the National Aeronautics and Space Administration Biodiversity and Ecological Forecasting program (NASA Grant NNX14AR62A), the Bureau of Ocean and Energy Management Ecosystem Studies program (BOEM award MC15AC00006) and NOAA in support of the Santa Barbara Channel Biodiversity Observation Network. S. Schroeter assisted with the multivariate analyses.

## Author contributions

D.R., L.W. and T.B. conceived the study and D.R. wrote the manuscript with assistance from A.R., R.M. and L.W. D.R., A.R. and S.H. were responsible for collecting and managing the site-based kelp forest time series data. D.R., S.H., A.R. and R.M. analysed the site-based kelp forest and coastal oceanography data and T.B. processed and analysed the satellite data of giant kelp biomass and SST. All authors discussed the results and commented on the manuscript.

## Additional information

**Competing financial interests:** The authors declare no competing financial interests.

