## [Peer Review File · Nature Communications]

Reviewers' Comments:

Reviewer #1 (Remarks to the Author):

- Summary of the key results, originality and interest

The authors present a very valuable long-term dataset, which includes metrics of kelp abundance and its associated ecological community as well as measures of the main environmental parameters known to generally influence kelp forests (temperature, nutrients, wave action).

The premise that underpins this study is that giant kelp forests are not as sensitive to warming as previously thought, and are therefore not suitable sentinel species that can provide early warning of climate change impacts in temperate reefs. This premise should certainly be of great interest to others in the field, as well as being of great relevance to managers.

- Data & methodology/ Appropriate use of statistics/ Conclusions

My main issue with this manuscript is that it lacks a quantitative assessment of the data. There are no appropriate formal statistical analyses of the main observed temporal patterns in kelp abundance or temperature or how these two may be related to one another, which are instead largely described qualitatively.

It's not clear to me how the authors mark the beginning of the 'warming' period to March 2013. Anomalies of up to +2 deg are normal through the data series. Is there a quantitative justification for starting in March 2013? Why not start a few months later, after the decline in temperature over autumn when temperatures were neutral/ slightly negative?

Overall, my general impression when looking at the data presented is that temperature in the study region has certainly increased markedly, especially since 2014, and kelp abundance has decreased overall, although with no dramatic losses associated with the most extreme recent warming.

Indeed one of the few temporal analyses presented (though not explained in the Methods - was this a simple regression?) does state this: "Overall, kelp declined during the 15-year record ($r^2 = 0.32$, $p = 0.03$), but this trend began in 2007, six years before the onset of anomalously warm conditions." Discounting a potential effect of temperature (and associated lower nutrients) on the overall decline of kelp does not make sense to me. Although the authors do present a longer-term temperature dataset (Fig 3b), this again is not analysed for temporal trends.

The observed decline in sessile invertebrates is again considered to not be linked to warming because decline began before the most recent high temperature anomalies. As per previous comment above, I don't think it makes sense to look at the last fifteen years in isolation (as the

authors do argue for kelp abundance patterns in L123). There's obviously a need to consider what is happening to this community in the context of a longer time-series, especially given how dynamic this system is (with great inter-annual variability).

I would also argue it makes sense to look at the relationship between sessile invertebrates and kelp abundance - i.e. how much of the decline in benthic invertebrates can be explained by the long term decline in kelp?

Wave action - this again deserves a more quantitative approach. What was the magnitude of storms that was associated with loss of kelp forests in the 20th Century? Also not sure that maximum wave height the best metric here.

Urchins - the authors record a very marked decline in these important herbivores (possibly due to temperature and disease). The authors argue that the mass mortalities of sea urchins have not led to corresponding increases in kelp. However, this may be because kelp is also independently being affected by temperature, i.e. this is a multi-stressor system that needs to be considered as such.

There's abundant literature that shows that resilience is greatly affected by multiple stressors. For example, kelp is less resilient to 'clearing' disturbances in warmer temperatures (e.g. Wernberg et al. 2010). It is quite likely that if urchin populations had been healthy (i.e. herbivory had been a stressor in the system), these kelp forests would have been affected by warming a lot more. In this particular case, it is arguably the lack of herbivory that may be providing some more resilience to kelp populations facing warming. While the authors argue that "The decline in sea urchins coincided with a decrease rather than an increase in giant kelp and little change in understory algae (Fig. 2a, b)" - I can't really see that in the data: a quantitative analysis of this relationship would again have helped.

- Suggested improvements

I think this study would benefit greatly from thorough and detailed analyses of the data to gain a more robust and rigorous understanding of what is happening as per comments above. All analyses are centred on comparing communities before and after the 'warming' but as per comment above I can't see a good justification for starting the 'warming' in March 2013, which may well affect the results of these analyses.

I also think a more in-depth consideration of the potential effects of a lack of herbivory as a stressor in the system in slowing down the effects of temperature on kelp is warranted.

Finally, I would also have liked to see some more context to this study, i.e. is there any emerging evidence from nearby regions that this recent warming has impacted other kelp forests?

Further comments:

L156 - More site information of the 9 forest sites would be valuable - e.g. include a map/ GPS details as supplementary material

L164 - Not clear about the need to include nitrate concentration data given it was modelled as a function of temperature alone (temperature data then sufficient?)

L168 - Chl a concentrations: what's the relative location of the 5 where Chl a was measured in relation to the 9 sites? - were these interspersed/ central? Add as icons to Map in supplementary material?

Figure 2 - Are these averaged for all 9 sites? why are there no error bars in this plot?!

Figure 3 - are these monthly anomalies? Need to specify

REFERENCES MENTIONED

Bennett, S., T. Wernberg, B. A. Joy, T. de Bettignies, and A. H. Campbell. 2015. Central and rear-edge populations can be equally vulnerable to warming. *Nature communications* 6.

Wernberg, T., M. S. Thomsen, F. Tuya, G. A. Kendrick, P. A. Staehr, and B. D. Toohey. 2010. Decreasing resilience of kelp beds along a latitudinal temperature gradient: potential implications for a warmer future. *Ecology Letters* 13:685-694.

Reviewer #2 (Remarks to the Author):

This paper reports the lack of effect of a recent warming event on giant kelp forests in California. The forest itself was unaffected but other species associated with the kelp both increased and declined. The responses are interesting both locally and globally, showing how coastal ecosystems than depend on foundation species can respond to extreme temperatures. The findings are used to challenge the assessment of the last IPCC report that kelp forests are declining in response to climate on the basis of evidence from kelp in Australia and Europe, and the idea that kelp might be a sentinel for climate change. While effectively refuting these propositions on the basis of recent changes in California, it must be observed that both these ideas are just "straw men" that almost any deeper consideration of climate change responses of the species involved would blow away. The IPCC report (Wong et al. 2014), for example, takes its evidence from declines in species near their equatorial range edges (*Saccorhiza polyschides* in Spain, for example), and so while it is true to say that kelp are in decline in these areas, the idea that kelp declines are expected globally in relation to climate is not supported. Few studies address potential increases in kelp at their cold border, where warming may allow increases. Likewise, the idea of sentinel species is an attractive one on the basis of the much reduced cost of tracking changes in a handful of conspicuous types, but without an a priori reason there is not a solid expectation for giant kelp to be any more sensitive to temperature than any other member of the kelp forest community. As it turns out, giant kelp in the Santa Barbara Channel is a non-responder, with the authors proposal that "Santa Barbara does not represent either the geographical or thermal limit for the species in the region (although it is in the warmer portion of the range" being the best explanation why. The complex interrelation of temperature and coastal

nutrients generated in that part of the world might work for those local populations of kelp ("Thus, we would expect giant kelp populations far from the range edge to be sensitive to large and sustained anomalies in temperature and nutrients such as those associated with the recent warming event.") but the results of this study don't support that idea either.

The results are well presented and easy to follow, with only small changes suggested here:

Fig. 1. SST anomalies are useful, but the long-term average SST for the area should also be given (around 16°C according to (Blanchette et al. 2006)). How does this value compare to other average SSTs throughout the range of *Macrocystis pyrifera*? Algaebase gives the species as present in Baja California where average temperatures must be much warmer.

Fig. 2. The grouping of species into sessile invertebrates, sea stars, sea urchins is only helpful for interpreting the broad changes in functional groups, but doesn't say anything about species identity. Is it possible to indicate the major contributing species to these changes? Likewise, it would be useful for the unfamiliar reader for the species in Supplementary Table 1 to be identified as belonging to one of these groups. It would also be good to know which of the species in this Supplementary Table 1 were cold-affinity or warm-affinity, to see whether the changes are consistent with climate.

Blanchette, C.A., Broitman, B.R. & Gaines, S.D., 2006. Intertidal community structure and oceanographic patterns around Santa Cruz Island, CA, USA. *Marine Biology*, 149(3), pp.689-701.

Wong, P.P. et al., 2014. Coastal systems and low-lying areas. In C. B. Field et al., eds. *Climate Change 2014: Impacts, Adaptation, and Vulnerability. Part A: Global and Sectoral Aspects. Contribution of Working Group II to the Fifth Assessment Report of the Intergovernmental Panel of Climate Change*. Cambridge, United Kingdom and New York, NY, USA: Cambridge University Press, pp. 361-409.

Reviewer #3 (Remarks to the Author):

In my opinion this is a very important paper given the international interest in detecting global change effects and the potential use of 'sentinel' species to do so. The Santa Barbara LTER is one of the few research programs with sufficient data to evaluate the utility of sentinel species in temperate waters. The comprehensive physical and biological data from the program pre and during an exceptional warming event clearly show that suggestions about suitable sentinel species based on limited data should be considered with caution. The paper should have a significant effect on future efforts to detect the biological effects of global change.

My only suggestion is that the authors put the temperature anomalies in the context of 'normal' maximum temperatures found in the Santa Barbara region since much of the prior literature is

about the maximum temperatures (and associated low nutrients) at which giant kelp deteriorates. This could be done by simply adding a sentence or two to the text.

AUTHOR'S RESPONSES IN ITALICS

Reviewer #1 (Remarks to the Author):

- Summary of the key results, originality and interest

The authors present a very valuable long-term dataset, which includes metrics of kelp abundance and its associated ecological community as well as measures of the main environmental parameters known to generally influence kelp forests (temperature, nutrients, wave action).

The premise that underpins this study is that giant kelp forests are not as sensitive to warming as previously thought, and are therefore not suitable sentinel species that can provide early warning of climate change impacts in temperate reefs. This premise should certainly be of great interest to others in the field, as well as being of great relevance to managers.

- Data & methodology/ Appropriate use of statistics/ Conclusions

My main issue with this manuscript is that it lacks a quantitative assessment of the data. There are no appropriate formal statistical analyses of the main observed temporal patterns in kelp abundance or temperature or how these two may be related to one another, which are instead largely described qualitatively.

We thought that the conclusions we drew in our earlier submission were relatively obvious and could be supported by the graphed data alone. We appreciate Reviewer 1 for bringing this concern to our attention. To provide more rigorous statistical support for our conclusions we analyzed our data using stepwise multiple regression to examine the amount of variation in each of the six ecological response variables shown in Fig. 3 that was explained by anomalies in bottom temperature, bottom nitrate and maximum significant wave height. We included year as an additional independent variable in these analyses to distinguish temporal trends that were independent of anomalous oceanographic conditions. The regression results are presented in Table 1 in the revised manuscript and are consistent with the qualitative assessment of the data that we presented in our earlier submission. The new analysis did reveal an unexpected positive relationship between sea urchin biomass and nitrate anomalies. One possible explanation for this result was that nitrate reflected favorable conditions for macroalgae, which are the primary food of sea urchins. To test this hypothesis we ran a stepwise multiple regression to examine whether kelp or understory algae explained any of the variation in sea urchin biomass; they did not, as neither variable met the minimum significance for entry into the model regardless of whether data for kelp and understory algae were derived from the same year as sea urchins or the previous year.

It's not clear to me how the authors mark the beginning of the 'warming' period to March 2013. Anomalies of up to +2 deg are normal through the data series. Is there a quantitative justification for starting in March 2013? Why not start a few months later, after the decline in temperature over autumn when temperatures were neutral/ slightly negative? *In our revised manuscript we applied the formal criteria for marine heatwaves developed by Hobday et al. (2016) to our data. We added a new figure (Fig. 1b) that uses these criteria to show the occurrence of heat waves in our times series. This analysis confirms this reviewer's suspicion that the anomalous warm period began in late December 2013 rather than March 2013. The analysis also clearly shows that heatwave days were nearly 20 times more common*

and heatwave events lasted three times longer in 2014 and 2015 compared to 2001-2013. Based on these results we changed the start of the warming period to late Dec 2013 (effectively January 2014 for all of our analyses). We developed a similar metric for low nitrate events (i.e., 21 continuous days of $< 1 \text{ } \mu\text{mol L}^{-1}$) using formalized criteria based on the known capacity for giant kelp to uptake and store nitrate. The results of this analysis (shown in new Fig. 1d) are consistent with those shown for heatwaves; the occurrence of low nitrate events was nearly eight times higher in 2014 – 2015 compared to 2001-2013.

Overall, my general impression when looking at the data presented is that temperature in the study region has certainly increased markedly, especially since 2014, and kelp abundance has decreased overall, although with no dramatic losses associated with the most extreme recent warming.

We agree with this general impression, which is one of the main conclusions of our study.

Indeed one of the few temporal analyses presented (though not explained in the Methods - was this a simple regression?) does state this: "Overall, kelp declined during the 15-year record ($r^2 = 0.32$, $p = 0.03$), but this trend began in 2007, six years before the onset of anomalously warm conditions." Discounting a potential effect of temperature (and associated lower nutrients) on the overall decline of kelp does not make sense to me. Although the authors do present a longer-term temperature dataset (Fig 3b), this again is not analysed for temporal trends.

We added the results of multiple regressions in Table 1 that included the analysis of temporal trends for all ecological response variables including giant kelp (see response to comment above). We have analyzed the longer term kelp data set for temporal trends (there was no trend; $r^2 < 0.02$, $p = 0.25$) but we chose not to report the results of these analyses in our paper because the purpose for presenting the longer term kelp data was to not examine long-term temporal trends, but rather simply to show that kelp biomass during the anomalously warm years of 2014-2015 was not drastically different from that observed in other years when temperatures were much cooler.

The observed decline in sessile invertebrates is again considered to not be linked to warming because decline began before the most recent high temperature anomalies. As per previous comment above, I don't think it makes sense to look at the last fifteen years in isolation (as the authors do argue for kelp abundance patterns in L123). There's obviously a need to consider what is happening to this community in the context of a longer time-series, especially given how dynamic this system is (with great inter-annual variability).

Because sessile invertebrates and other components of the kelp forest communities cannot be sampled remotely by satellites we do not have a longer time series of their biomass that is similar to that presented for giant kelp. Nonetheless, our 15-year time series of community data is longer than the vast majority of published studies and is sufficient for determining whether sessile invertebrates and other components of the community showed a large response to the anomalous conditions in 2014-2015. This is evidenced by the results from multiple regression analyses, which clearly show a significant temporal decline during the 15-year study period that was unrelated to positive anomalies in temperature. Moreover, many of the components of the kelp forest community, including sessile invertebrates, do not show the same high interannual variability as giant kelp and thus do not require as long of a time series to detect whether there was a directional response to warming as was shown for sea stars (but not for sessile

invertebrates).

I would also argue it makes sense to look at the relationship between sessile invertebrates and kelp abundance - i.e. how much of the decline in benthic invertebrates can be explained by the long term decline in kelp?

Our system, like that of many dominated by kelps, is complex with a myriad of physical and biological processes interacting to affect species composition and community dynamics. We have previously published several papers on this topic including papers that specifically investigated interactions between giant kelp and benthic sessile invertebrates (e.g., Arkema et al. 2009 Ecology, 90: 3126-3137; Byrnes et al. 2011 Global Change Biology 17: 2513-2524; Miller et al. 2015 Oecologia 179:1199-1209). Rather than to attempt to explain the causes for all the community dynamics observed in our time series, our goal in this paper is to examine the value of giant kelp as a sentinel species for climate change and to highlight the fact that the negative effects of high temperature and low nutrients on giant kelp were not nearly as dominant of a factor as others have suggested or as we ourselves expected before the recent warming event. Consequently, we have chosen not to include results of analyses that are tangential to this primary goal.

Wave action - this again deserves a more quantitative approach. What was the magnitude of storms that was associated with loss of kelp forests in the 20th Century? Also not sure that maximum wave height the best metric here.

To address this concern we compiled longer term records of swell height (1980-2015) obtained from hindcast data modeled from an offshore buoy in the Santa Barbara Channel. These data show that monthly anomalies in maximum significant wave height were 2-5 times higher during the warm years of the 1982-83 and 1997-1998 El Niños (new Fig 2a). We used a shorter record of maximum significant wave height modeled for each of our nine study sites to show that largest waves during our 15-year study actually occurred during the cool years (2006 and 2008) rather than the warm years of 2014 and 2015. We chose to use maximum significant wave height as a metric for swell height because we have previously shown that it is a good predictor of the loss of kelp biomass (we added a statement to this effect in the methods with the relevant citations).

Urchins - the authors record a very marked decline in these important herbivores (possibly due to temperature and disease). The authors argue that the mass mortalities of sea urchins have not led to corresponding increases in kelp. However, this may be because kelp is also independently being affected by temperature, i.e. this is a multi-stressor system that needs to be considered as such.

We added the results from multiple regression analyses that show year-to-year fluctuations in sea urchin biomass were unrelated to the biomass of giant kelp and understory algae. Considering kelp forests as a multi-stressor system is an excellent point, and we completely agree about the complex multi-causal nature of this system. It is this very complexity that makes predicting the effects of climate change on kelp so challenging. We have added text in the paragraph on sea urchins noting this point.

There's abundant literature that shows that resilience is greatly affected by multiple stressors. For example, kelp is less resilient to 'clearing' disturbances in warmer temperatures (e.g. Wernberg et al. 2010). It is quite likely that if urchin populations had been healthy (i.e. herbivory had been a

stressor in the system), these kelp forests would have been affected by warming a lot more. In this particular case, it is arguably the lack of herbivory that may be providing some more resilience to kelp populations facing warming. While the authors argue that "The decline in sea urchins coincided with a decrease rather than an increase in giant kelp and little change in understory algae (Fig. 2a, b)" - I can't really see that in the data: a quantitative analysis of this relationship would again have helped.

The urchin story has changed a bit based on the multiple regression results and we modified the text accordingly. Specifically we no longer argue that "The decline in sea urchins coincided with a decrease rather than an increase in giant kelp and little change in understory algae". Instead we refer to the results of the multiple regression analyses and report that changes in sea urchin biomass were unrelated to the biomass of giant kelp and understory algae. We wholeheartedly agree that the dynamics of kelp forest communities are affected by multiple stressors. To emphasize this point we added a statement acknowledging the complexities of biotic and abiotic interactions in kelp forests (with a citation to Wernberg et al. 2010) as a possible reason for the poor relationship between sea urchins and their primary food sources.

- Suggested improvements

I think this study would benefit greatly from thorough and detailed analyses of the data to gain a more robust and rigorous understanding of what is happening as per comments above. All analyses are centred on comparing communities before and after the 'warming' but as per comment above I can't see a good justification for starting the 'warming' in March 2013, which may well affect the results of these analyses.

We added additional analyses as noted above to provide a more quantitative basis for our conclusions. This resulted in using January 2014 as the beginning of the warming period in our analyses, which did not change any of our conclusions concerning the response of the giant kelp and its associated community to the anomalous warming.

I also think a more in-depth consideration of the potential effects of a lack of herbivory as a stressor in the system in slowing down the effects of temperature on kelp is warranted.

We added a statement concerning the complexities of biotic and abiotic interactions in kelp dominated systems as a possible reason for the poor correlation that we observed between sea urchins and the primary food (giant kelp and understory algae). The inability of kelp and understory algae to explain the relatively high variability observed in sea urchins during the 15-year time series suggests that sea urchins are not a major driver of algal biomass at regional scales (we have documented this in previous work: Reed et al. 2011, Ecology 92:2108-2116; Bell et al. 2015, J. Biogeogr. 42, 2010–2021). Ultimately, separating out any possible effect of decreased herbivory on kelp recovery at smaller scales would require manipulative experiments, and we have avoided speculation and conclusions that reach beyond our results.

Finally, I would also have liked to see some more context to this study, i.e. is there any emerging evidence from nearby regions that this recent warming has impacted other kelp forests?

We added long-term (32 years) data on giant kelp biomass and sea surface temperature from Los Angeles and San Diego counties. Data from these more southern and warmer kelp forests show remarkably similar patterns to those obtained for Santa Barbara in that large anomalous declines in kelp biomass did not accompany the unprecedented warming in 2014- 2015.

Further comments:

L156 - More site information of the 9 forest sites would be valuable - e.g. include a map/ GPS details as supplementary material

We added a map showing the locations and Lat / Long coordinates for each of our nine study sites in supplementary material (Supplementary Fig. 4).

L164 - Not clear about the need to include nitrate concentration data given it was modelled as a function of temperature alone (temperature data then sufficient?)

The relationship between nitrate and temperature is non-linear and the collinearity between the two variables was low in all analyses. Consequently, we think that there is much value in presenting the nitrate data derived from temperature and have chosen to retain them.

L168 - Chl a concentrations: what's the relative location of the 5 where Chl a was measured in relation to the 9 sites? - were these interspersed/ central? Add as icons to Map in supplementary material?

We omitted the Chl a data from the figure showing oceanographic anomalies in our revised manuscript because they were tangential to our paper. We replaced these data with additional plots of temperature, nitrate and swell height since these variables were the focus of our analyses.

Figure 2 - Are these averaged for all 9 sites? why are there no error bars in this plot?!

We clarified in the figure legend that the values in the plots represent means averaged over all nine sites. We chose not to include error bars for clarity since variation among sites is not relevant to our conclusions. Instead the focus of our study is on regional responses to warming and the mean of the nine sites is our best estimate of the region.

Figure 3 - are these monthly anomalies? Need to specify

Clarification made.

REFERENCES MENTIONED

Bennett, S., T. Wernberg, B. A. Joy, T. de Bettignies, and A. H. Campbell. 2015. Central and rear-edge populations can be equally vulnerable to warming. *Nature communications* 6.

Wernberg, T., M. S. Thomsen, F. Tuya, G. A. Kendrick, P. A. Staehr, and B. D. Toohey. 2010. Decreasing resilience of kelp beds along a latitudinal temperature gradient: potential implications for a warmer future. *Ecology Letters* 13:685-694.

Both of these references were added to the revised manuscript.

Reviewer #2 (Remarks to the Author):

This paper reports the lack of effect of a recent warming event on giant kelp forests in California.

The forest itself was unaffected but other species associated with the kelp both increased and declined. The responses are interesting both locally and globally, showing how coastal ecosystems than depend on foundation species can respond to extreme temperatures. The findings are used to challenge the assessment of the last IPCC report that kelp forests are declining in response to climate on the basis of evidence from kelp in Australia and Europe, and the idea that kelp might be a sentinel for climate change. While effectively refuting these propositions on the basis of recent changes in California, it must be observed that both these ideas are just "straw men" that almost any deeper consideration of climate change responses of the species involved would blow away. The IPCC report (Wong et al. 2014), for example, takes its evidence from declines in species near their equatorial range edges (*Saccorhiza polyschides* in Spain, for example), and so while it is true to say that kelp are in decline in these areas, the idea that kelp declines are expected globally in relation to climate is not supported. Few studies address potential increases in kelp at their cold border, where warming may allow increases. *We modified the Abstract and Introduction of our paper to clarify that our results challenge general assumptions concerning the high vulnerability of kelps to extreme warming rather than specific claims by the IPCC report that kelp forests are generally declining in response to changes in climate.*

Likewise, the idea of sentinel species is an attractive one on the basis of the much reduced cost of tracking changes in a handful of conspicuous types, but without an a priori reason there is not a solid expectation for giant kelp to be any more sensitive to temperature than any other member of the kelp forest community. As it turns out, giant kelp in the Santa Barbara Channel is a non-responder, with the authors proposal that "Santa Barbara does not represent either the geographical or thermal limit for the species in the region (although it is in the warmer portion of the range" being the best explanation why. The complex interrelation of temperature and coastal nutrients generated in that part of the world might work for those local populations of kelp ("Thus, we would expect giant kelp populations far from the range edge to be sensitive to large and sustained anomalies in temperature and nutrients such as those associated with the recent warming event.") but the results of this study don't support that idea either. *We edited the text to better emphasize the physiological and demographic characteristics of giant kelp make it an obvious a priori candidate for a sentinel species. Specifically its rapid growth and high biomass turnover coupled with its relatively high nitrogen demand and limited capacity for nitrogen storage provide substantial a priori reasons to expect it to display a rapid response to prolonged conditions of extreme warming that coincide with low nutrients. Moreover, the literature is replete with expectations of its susceptibility to such conditions, which was our expectation prior to the 2014-2015 warming event.*

The results are well presented and easy to follow, with only small changes suggested here:

Fig. 1. SST anomalies are useful, but the long-term average SST for the area should also be given (around 16°C according to (Blanchette et al. 2006)). How does this value compare to other average SSTs throughout the range of *Macrocystis pyrifera*? Algaebase gives the species as present in Baja California where average temperatures must be much warmer. *We added a figure showing the monthly mean SST for SB, LA and SD averaged over 1980-2015 (Supplementary Figure 3).*

Fig. 2. The grouping of species into sessile invertebrates, sea stars, sea urchins is only helpful for interpreting the broad changes in functional groups, but doesn't say anything about species identity. Is it possible to indicate the major contributing species to these changes? Likewise, it would be useful for the unfamiliar reader for the species in Supplementary Table 1 to be identified as belonging to one of these groups. It would also be good to know which of the species in this Supplementary Table 1 were cold-affinity or warm-affinity, to see whether the changes are consistent with climate.

We added a list of all the species in each function group as Table 2 in the supplementary material. All the species listed in Supplementary Table 1 are sessile invertebrates, which is stated in the table legend. The taxonomic order of each species can be obtained by cross reference with Supplementary Table 2. Information on the abundance, geographic distributions and natural history of most of the species that we sampled is not sufficient for us to assign them to having a cold or warm affinity.

Blanchette, C.A., Broitman, B.R. & Gaines, S.D., 2006. Intertidal community structure and oceanographic patterns around Santa Cruz Island, CA, USA. *Marine Biology*, 149(3), pp.689-701.

Wong, P.P. et al., 2014. Coastal systems and low-lying areas. In C. B. Field et al., eds. *Climate Change 2014: Impacts, Adaptation, and Vulnerability. Part A: Global and Sectoral Aspects. Contribution of Working Group II to the Fifth Assessment Report of the Intergovernmental Panel of Climate Change*. Cambridge, United Kingdom and New York, NY, USA: Cambridge University Press, pp. 361-409.

Reviewer #3 (Remarks to the Author):

In my opinion this is a very important paper given the international interest in detecting global change effects and the potential use of 'sentinel' species to do so. The Santa Barbara LTER is one of the few research programs with sufficient data to evaluate the utility of sentinel species in temperate waters. The comprehensive physical and biological data from the program pre and during an exceptional warming event clearly show that suggestions about suitable sentinel species based on limited data should be considered with caution. The paper should have a significant effect on future efforts to detect the biological effects of global change.

My only suggestion is that the authors put the temperature anomalies in the context of 'normal' maximum temperatures found in the Santa Barbara region since much of the prior literature is about the maximum temperatures (and associated low nutrients) at which giant kelp deteriorates. This could be done by simply adding a sentence or two to the text.

We added a figure in the supplementary material showing the monthly mean SST for SB, LA and SD averaged over 1980-2015 (Supplementary Figure 3).

Reviewers' Comments:

Reviewer #1 (Remarks to the Author):

The MS is much improved after the addition of (i) new quantitative analyses of the data, (ii) formal criteria for marine heatwaves, and (iii) new wave height data. There are a number of issues that I think still need resolving before publication. In particular, I think adding an estimate of variance among sites within years in Figure 3 is essential (see below). Some other issues are noted below:

L73 – Insert marine heatwave criteria used here

The threshold significance level of 0.15 (indicated in Table 1 (L417) needs to be justified.

The use of asterisks in Table 1 to denote non-significance is confusing, as these are generally used to denote significance in most publications. Substitute by 'n.s.'?

With regards to the PERMANOVA + SIMPER analyses to compare community structure of algae/ invertebrates/ fishes: something I did not pick up in the original review but that I think may be an issue relates to the bias due to the unbalanced dataset. There are only 2 years post warming vs. 13 years pre warming –the pre-warming period is therefore much better represented. One option around this would be to pick two random years pre warming and compare them to the two years post-warming. At the very least the limitations of such a skewed approach need to be fully acknowledged.

Figure 1. The use of blue and red needs to be explained in the Figure legend.

Figure 2. Could be useful to mark with arrows previous ENSO events?

Figure 3. The authors argue that they choose not to present error bars for clarity because they are interested in regional responses to warming and variation among sites is not relevant. However, including an estimate of the variance is absolutely essential to interpret temporal trends if variance varies between years. E.g. imagine a situation where there is a lot of variation among sites but only towards the end of the temporal series and not in the first few years – even though the means may show a decline, the temporal trend may not be significant due to differences in variance.

I agree that adding error bars onto Figure 3 will reduce the clarity – an alternative may be to use a generalized linear mixed effects model (GLMM), e.g. with kelp abundance as the response

variable, year as a continuous predictor and sites as the random intercept, and representing standard errors of model fits as shading in the figure. Separate GLMMs could then be used to predict effects of environmental drivers on kelp (or any other variable), with kelp as the response variable, Temperature, Nitrate, Max Hs as fixed predictors, with again sites nested in year as a random intercept and year incorporated as a random effect to include temporal auto-correlation.

Reviewer #2 (Remarks to the Author):

The amended abstract and Introduction do help to change the impression that the paper is intended to challenge the specific claims of the last IPCC report that kelps are declining in response to warming. My original point was the failure of the species to respond might not be surprising, given that vulnerability to high temperatures might not be expected in a particular location for a species that experienced extremes similar to temperatures regularly experienced elsewhere in the geographical range. The authors point out that the combination of nutrient changes combined with high temperatures is indeed a threat to this species, and I defer to them on this point. The extra information on actual temperatures is a significant improvement, and the addition of data on changes in giant kelp at two other locations strongly reinforces the conclusion of an unexpected non-response to recent high temperatures.

Reviewer #3 (Remarks to the Author):

I thought the original submitted ms. was excellent and addressed a very important issue regarding assessment of the effects of ocean climate change. I had only one suggestion for revision. The authors well revised relative to my suggestion and, in my opinion, those of other reviewers. An excellent, timely, and important paper.

One small detail: The "of" before sentinel in the abstract should be removed.

Responses to reviewer comments (in *italics* below each comment)

Reviewer #1 (Remarks to the Author):

The MS is much improved after the addition of (i) new quantitative analyses of the data, (ii) formal criteria for marine heatwaves, and (iii) new wave height data. There are a number of issues that I think still need resolving before publication. In particular, I think adding an estimate of variance among sites within years in Figure 3 is essential (see below). Some other issues are noted below:

L73 – Insert marine heatwave criteria used here

Response: In our earlier submission we included a brief description of the criteria used for marine heatwaves in the Methods (with a reference to Hobday et al. 2016 who developed these criteria). As suggested we inserted the description of the marine heatwave criteria to the recommended location of the text and deleted it from the Methods.

The threshold significance level of 0.15 (indicated in Table 1 (L417) needs to be justified.

Response: We used SAS ver 9.4 for the stepwise linear regressions. We were most interested in choosing a regression model that provided the best predictive power for each response variable. SAS recommends significance values between 0.1-0.25 in such cases to guard against estimating more parameters than can be reliably estimated with the given sample size. SAS's default threshold level of significance for entry into a stepwise regression model is 0.15, which is the level that we used. We added a reference to SAS (ver9.4, SAS Institute Inc., Cary, NC, USA) in the Methods when describing the stepwise regression approach.

The use of asterisks in Table 1 to denote non-significance is confusing, as these are generally used to denote significance in most publications. Substitute by 'n.s.'?

Response: We agree the use of asterisks in Table 1 is confusing and have replaced them with a double dash (--). We chose not to replace the asterisks with n.s. because the asterisks denoted that the variable was not included in the model (as opposed to a variable being included in the model but not significant).

With regards to the PERMANOVA + SIMPER analyses to compare community structure of algae/ invertebrates/ fishes: something I did not pick up in the original review but that I think may be an issue relates to the bias due to the unbalanced dataset. There are only 2 years post warming vs. 13 years pre warming –the pre-warming period is therefore much better represented. One option around this would be to pick two random years pre warming and compare them to the two years post-warming. At the very least the limitations of such a skewed approach need to be fully acknowledged.

Response: The consequences for unbalanced designs for a one-way PERMANOVA such as ours are not problematic (Anderson et al. 2008). Concerns pertaining to unbalanced designs using

PERMANOVA arise when there is more than one factor in the design. Such concerns are typically treated using a Type III sums of squares. The default for PERMANOVA + using PRIMER is a Type III sums of squares, which is what we used in our analyses to compare community structure. We added a statement to the Methods that we used a Type III sums of squares when using PERMANOVA.

Figure 1. The use of blue and red needs to be explained in the Figure legend.

Response: We changed the text pertaining to the use of red in the legend of Fig 1 to match that of Fig. 2, which more thoroughly explains that red indicates the anomalously warm years of 2014-2015.

Figure 2. Could be useful to mark with arrows previous ENSO events?

Response: arrows were added to Fig 2a to denote the 1982-83 and 1997-98 ENSO events.

Figure 3. The authors argue that they choose not to present error bars for clarity because they are interested in regional responses to warming and variation among sites is not relevant. However, including an estimate of the variance is absolutely essential to interpret temporal trends if variance varies between years. E.g. imagine a situation where there is a lot of variation among sites but only towards the end of the temporal series and not in the first few years – even though the means may show a decline, the temporal trend may not be significant due to differences in variance.

I agree that adding error bars onto Figure 3 will reduce the clarity – an alternative may be to use a generalized linear mixed effects model (GLMM), e.g. with kelp abundance as the response variable, year as a continuous predictor and sites as the random intercept, and representing standard errors of model fits as shading in the figure. Separate GLMMs could then be used to predict effects of environmental drivers on kelp (or any other variable), with kelp as the response variable, Temperature, Nitrate, Max Hs as fixed predictors, with again sites nested in year as a random intercept and year incorporated as a random effect to include temporal auto-correlation.

Response: We agree that GLMMs would be a better approach if the contribution of random site effects was a focus of our study. Instead, we used stepwise multiple regression to evaluate temporal trends in the regional biomass of each functional group using a single regional mean value of biomass for each year. Thus there is no variance associated with site in the regression models, and including error bars on Figure 3 would not be consistent with how the data were analyzed or with the questions that the analyses were used to address. However, it is clear to us now how this could be confusing, and to clarify this issue we replotted Figure 3 as scatter plots (with regression lines for significant regression models) rather than line graphs to emphasize the presence (or absence) of temporal trends in biomass. We added a supplemental figure (Supplemental Figure 2) of scatter plots (and significant regression lines where appropriate) to show the relationship between biomass of each functional group and bottom temperature anomaly (Supplemental Figure 2).

Reviewers' Comments:

Reviewer #1 (Remarks to the Author):

There are two issues that still need to be revised prior to publication, in my opinion.

1. Regression analysis and figure 3: ecosystem responses to extreme warming by kelp forests

In their recent reply letter, the authors explain that they used a single regional mean per year in their stepwise multiple regression to evaluate temporal trends on the biomass of kelp, algae, sessile invertebrates, etc. They argue that GLMMs were not used because site effects were not a focus of the study.

I disagree with this argument and think that the appropriate analysis needs to include the variability among sites. Including 'site' as a random factor in a generalized mixed effects model is not done because of an interest in examining site effects, but to account for/ incorporate this source of spatial variability.

As per my previous review, I recommend using a generalized linear mixed effects model (GLMM), with kelp abundance as the response variable, year as a continuous predictor and sites as the random intercept, and representing standard errors of model fits as shading in the figure. Separate GLMMs could then be used to predict effects of environmental drivers on kelp (or any other variable), with kelp as the response variable, Temperature, Nitrate, Max Hs as fixed predictors, with again sites nested in year as a random intercept and year incorporated as a random effect to include temporal auto-correlation.

2. With regards to the unbalanced dataset in the PERMANOVA design, whereby the authors compare 13 years pre-warming and 2 years post-warming. While PERMANOVA may be able to deal with some unbalanced designs, this depends on the dispersion among the data points in the different groups. I refer the authors to a paper from Anderson & Walsh (2013), where they examine the effects of heterogeneity on PERMANOVA analyses, which they find really matters in unbalanced designs. In this particular example, where there appears to be a temporal pattern of change in important variables, e.g. a decline in kelp, it seems to me that this before/ after heatwave approach is missing important information.

Anderson, M. J., and D. C. I. Walsh. 2013. PERMANOVA, ANOSIM, and the Mantel test in the face of heterogeneous dispersions: What null hypothesis are you testing? *Ecological Monographs* 83:557-574.

Responses to reviewer comments (in *italics* below each comment)

Reviewer #1 (Remarks to the Author):

There are two issues that still need to be revised prior to publication, in my opinion.

1. Regression analysis and figure 3: ecosystem responses to extreme warming by kelp forests

In their recent reply letter, the authors explain that they used a single regional mean per year in their stepwise multiple regression to evaluate temporal trends on the biomass of kelp, algae, sessile invertebrates, etc. They argue that GLMMs were not used because site effects were not a focus of the study.

I disagree with this argument and think that the appropriate analysis needs to include the variability among sites. Including 'site' as a random factor in a generalized mixed effects model is not done because of an interest in examining site effects, but to account for/ incorporate this source of spatial variability.

As per my previous review, I recommend using a generalized linear mixed effects model (GLMM), with kelp abundance as the response variable, year as a continuous predictor and sites as the random intercept, and representing standard errors of model fits as shading in the figure. Separate GLMMs could then be used to predict effects of environmental drivers on kelp (or any other variable), with kelp as the response variable, Temperature, Nitrate, Max Hs as fixed predictors, with again sites nested in year as a random intercept and year incorporated as a random effect to include temporal auto-correlation.

Response- We recognize that including 'site' as a random factor in a generalized mixed effects model is done to account for/ incorporate this source of spatial variability. We chose not to analyze our data using this approach because we were not interested in the contribution of random site effects. Instead, the primary focus of our paper (and the purpose of the analyses within it) were on evaluating regional scale responses by kelp forest communities to a large-scale warming event. Consequently, our statistical models used regional means averaged across the nine sites rather than estimates of the mean and variance of variables from time series of individual sites (such as the regional mean and variance in the relationship between local kelp biomass and local temperature that would be derived from a mixed effects model). We believe that stepwise multiple regression is appropriate given our question and provides a simple and intuitive explanation for the patterns that we observed. We edited the manuscript to clarify the focus of our analyses and the rationale for the statistical approach that we used.

2. With regards to the unbalanced dataset in the PERMANOVA design, whereby the authors compare 13 years pre-warming and 2 years post-warming. While PERMANOVA may be able to deal with some unbalanced designs, this depends on the dispersion among the data points in the different groups. I refer the authors to a paper from Anderson & Walsh (2013), where they examine the effects of heterogeneity on PERMANOVA analyses, which they find really matters in unbalanced designs. In this particular example, where there appears to be a temporal pattern of change in important variables, e.g. a decline in kelp, it seems to me that this before/ after heatwave approach is missing important information.

Response: We appreciate the reviewer alerting us to the paper by Anderson and Walsh and the concerns with using PERMANOVA for unbalanced designs in the face of heterogeneity. We added a sentence to the Methods noting the sensitivity of PERMANOVA to heterogeneity in dispersions among groups in the case of unbalance designs. We direct the reader to the MDS plots and SIMPER in the text for assessing differences in species composition between the cool and warm periods and note in the Methods that the statistical significance of these differences as determined by PERMANOVA should be viewed with caution.